# Characterization of Absorption Losses and Transient Thermo-Optic Effects in a High-Power Laser System

**Lukasz Gorajek, Przemyslaw Gontar, Jan Jabczynski \*, Jozef Firak, Marek Stefaniak, Miroslaw Dabrowski, Tomasz Orzanowski, Piotr Trzaskawka, Tomasz Sosnowski, Krzysztof Firmanty, Marcin Miczuga, Jaroslaw Barela and Krzysztof Kopczynski**

Institute of Optoelectronics, Military University of Technology, ul gen. S. Kaliskiego 2, 00-908 Warsaw, Poland; lukasz.gorajek@wat.edu.pl (L.G.); przemyslaw.gontar@wat.edu.pl (P.G.); jozef.firak@wat.edu.pl (J.F.); marek.stefaniak@wat.edu.pl (M.S.); miroslaw.dabrowski@wat.edu.pl (M.D.); tomasz.orzanowski@wat.edu.pl (T.O.); piotr.trzaskawka@wat.edu.pl (P.T.); tomasz.sosnowski@wat.edu.pl (T.S.); krzysztof.firmanty@wat.edu.pl (K.F.); marcin.miczuga@wat.edu.pl (M.M.); jaroslaw.barela@wat.edu.pl (J.B.); krzysztof.kopczynski@wat.edu.pl (K.K.)

**\*** Correspondence: jan.jabczynski@wat.edu.pl; Tel.: +48-261-839-617

**Abstract:** (1) Background: The modeling, characterization, and mitigation of transient lasers, thermal stress, and thermo-optic effects (TOEs) occurring inside high energy lasers have become hot research topics in laser physics over the past few decades. The physical sources of TOEs are the un-avoidable residual absorption and scattering in the volume and on the surface of passive and active laser elements. Therefore, it is necessary to characterize and mitigate these effects in real laser systems under high-power operations. (2) Methods: The laboratory setup comprised a 10-kW continuous wave laser source with a changeable beam diameter, and dynamic registration of the transient temperature profiles was applied using an infrared camera. Modeling using COMSOL Multiphysics enabled matching of the surface and volume absorption coefficients to the experimental data of the temperature profiles. The beam quality was estimated from the known optical path differences (OPDs) occurring within the examined sample. (3) Results: The absorption loss coefficients of dielectric coatings were determined for the evaluation of several coating technologies. Additionally, OPDs for typical transmissive and reflective elements were determined. (4) Conclusions: The idea of dynamic self-compensation of transient TOEs using a tailored design of the considered transmissive and reflecting elements is proposed.

**Keywords:** high-power lasers; laser weapon; thermal optics; optical technology; absorption losses; thermal imaging; laser optics

## 1. Introduction

High energy lasers (HELs) with an average power of 100 kW are required in the fields of science, technology, and the military [1–8]. In the case of laser weapon (LaW) applications, such systems do not operate as steady-state, continuous-wave (CW) systems, as the required interaction time with a target is only a few seconds. The transient laser, thermal stress, and thermo-optic effects (TOEs) occurring inside a HEL and laser beam-forming tracking have to be characterized [9,10] and compensated. The main physical sources of TOEs are the un-avoidable residual absorption and scattering in the volume and on the surface of dielectric coatings of passive and active laser elements, which have been theoretically and experimentally studied for several decades (see, e.g., [11–13]). Therefore, the characterization and mitigation of these effects are required in real laser systems operating under a high power [9]. Furthermore, developing a technology for optical elements with such applications remains a great

challenge [14–16]. Therefore, understanding the effective absorption occurring inside laser elements in real, high-power systems is crucial for validating the optical technology.

Last year, we developed a laboratory model of a laser effector based on a commercial CW 10-kW fiber laser operating nominally in near single mode [6]. The serious problems associated with the performance of such a system, evidencing the dependence on the duty cycle and worsening of the average laser power beam quality [17,18], were identified. Therefore, we decided to examine, step-by-step, the elements and subsystems of this laser system. The optical elements deployed in this system, all of which are made of fused silica, were investigated in this study. This material was mainly selected because of its lowest available volume absorption (<3 ppm/cm according to vendor information) and very low thermal expansion coefficient (~5 × 10$^{-7}$ 1/K), resulting in a high thermal shock parameter required for high power densities and heat load operations. The laboratory setup, method, and results of the experiment of several optical elements are presented in Section 2. An analysis of the experiments via modeling in COMSOL Multiphysics enabled us to determine the surface and volume absorption coefficients of the examined laser elements. Furthermore, with the known 3D temperature distributions and surface deformation of a sample, the transient optical path difference (OPD) and the resulting beam quality reduction could be calculated.

The conclusions drawn from this analysis lead us to propose the concept of dynamic self-compensation of transient TOEs in laser optics tracking. This concept has been analyzed in several studies over the last 20 years, and the most challenging problem encountered in these studies is associated with the compensation of TOEs in the Laser Interferometry Gravitational-Wave Observatory (LIGO) interferometers [13,16,19–22].

Herein, we attempt to address the aforementioned issue by specifically designing transmissive and reflective elements made of the same optical medium (high-quality fused silica in our case). The analysis and preliminary numerical verification of such a concept based on our experimental results are presented in Section 3.

## 2. Experiments

In the examined laser system, we used two kinds of components—one part supplied by an external vendor and the other manufactured by our team (dichroic mirrors and lenses). We intended to validate both groups of optical elements for our specific purpose, so the information on temperature increases under a real high-power operation was of critical importance. Moreover, by applying the COMSOL model, we intended to estimate how these temperature profiles impact the beam quality of the overall optical train.

The state of the art in absorption loss measurements is a sensitivity below 1 ppm [12,13,16]. However, for the testing of high-power optics destined for LaW applications, the loss measurement sensitivity of 1–5 ppm seems to be satisfactory [15]. In our case, it was more important to validate the optics in the real laser system and characterize transient TOEs occurring during the typical exposition durations of 10–30 s. To achieve this goal, we used a method based on the comparison of dynamically registered 2D temperature maps, via an infrared camera (IRC), with a model of a sample implemented in COMSOL Multiphysics. The experimental setup is shown in Figure 1.

Following the approach in [14], we preliminary assessed the thermal response time $\tau_{TR}$ of the examined samples according to the following formula:

$$\tau_{TR} = (2l/\pi)^2 \kappa^{-1}, \tag{1}$$

where $l$ is the thickness of the laser element and $\kappa$ is the thermal diffusivity ($\kappa$ = 0.823 mm$^2$/s for fused silica).

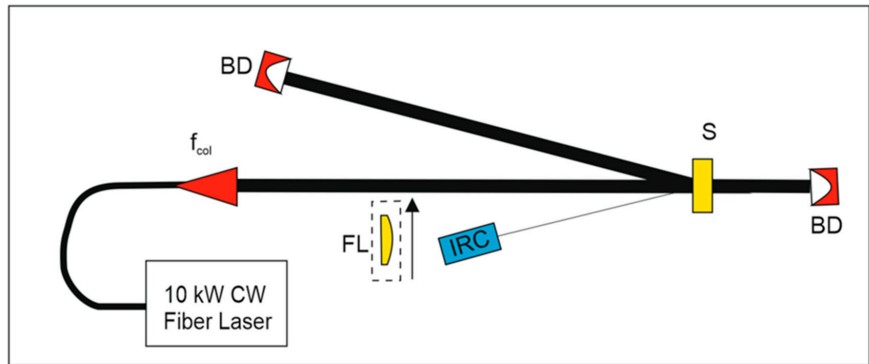

**Figure 1.** Experimental setup for the characterization of transient thermo-optic effects (TOEs): IRC: infrared camera; S: tested optical element; BD: beam dumper; FL: additional focusing lens $f$ = 11 m; and $f_{col}$: fiber collimator.

A high-power laser beam of 9.5 kW and diameter $2w_g$ = 19.6 mm were incident on an examined sample, which was slightly tilted to reflect the laser beam to the beam dumper. The thermal image of the examined surface under the incidence angle of ~5° was registered by means of an SC5600 IR camera with a nominal temperature resolution of 0.025 K. However, we estimated that the realistic temperature uncertainty was 0.2–0.3 K, which corresponds to our specific experimental conditions (incident power of 10 kW with a beam diameter of less than 2 cm), with uncertainty in absorption loss estimations of 2 ppm.

The exposition time for each sample (excluding $DCH_{2*}$) was 60 s. Afterwards, the laser operation was stopped. The exposition time was comparable to or shorter than the thermal response time for each sample (Table 1); thus, we assumed that the examined sample operates in the so-called 'pre-diffusion phase'. In the experiments, the laser beam diameter was more than two times smaller than the sample diameter, so truncation losses and heating of the mechanical mounts can be neglected.

**Table 1.** Measurement results: $\phi$: sample diameter; $l$: thickness of the laser element; $\tau_{TR}$: thermal response time; $t_{exp}$: exposition time, $\Delta T$: temperature increase; $A_{surf}$: surface absorption loss coefficient; and $\alpha$: volume absorption coefficient. $HR_1$ (technology No. 1), $HR_2$ (technology No. 2), and $HR_3$ (technology No. 2) are highly reflective mirrors specified for normal incidence. $DCH_1$ (technology No. 3) and $DCH_{2*}$- (technology No. 4) are dichroic mirrors specified for a 30° incidence. In the case of $DCH_{2*}$, because of the drastic increase in temperature, the exposition time was shortened to 18 s. $L_1$: lens with AR coatings (technology No. 5), and $S_4$: substrate without coatings. Here, we assumed residual surface losses of 5 ppm.

| Parameter | $HR_1$ | $HR_2$ | $HR_3$ | $DCH_1$ | $DCH_{2*}$ | $L_1$ | $S_4$ |
|---|---|---|---|---|---|---|---|
| $\phi$ (mm) | 50 | 50 | 100 | 75 | 75 | 140 | 140 |
| $l$ (mm) | 10 | 10 | 15 | 15 | 15 | 25 | 25 |
| $\tau_{TR}$ (s) | 49.2 | 49.2 | 110 | 110 | 110 | 307 | 307 |
| $t_{exp}$ (s) | 60 | 60 | 60 | 60 | 18 | 80 | 60 |
| $\Delta T$ @ $2w_g$ = 19.6 mm | 3.3 | 10.3 | 8.7 | 11.4 | 90.4 | 2.3 | 1.5 |
| $A_{surf}$ (ppm) | 16.3 | 50.5 | 44 | 57 | 670 | 8.5 | 5 |
| $\alpha$ (ppm/cm) | N/A | N/A | N/A | N/A | N/A | 5 | 5 |
| $\Delta T$ @ $2w_g$ = 14 mm | 5.8 | 14.2 | 12.6 | 18.1 | N/A | N/A | N/A |
| $A_{surf}$ (ppm) | 18.5 | 46 | 42 | 60 | N/A | N/A | N/A |

For each exposition, 2D thermal images were registered and the maximal temperature was found for the time interval of 0.5 s (Figures 2 and 3). To verify the measurement method, we used an additional focusing lens (focal length $f$ = 11 m), and after thermal stabilization, processed the second exposition with a slightly decreased beam diameter of $2w_g$ = 14 mm. The 2D temperature distributions were recorded and compared to the results of the model in COMSOL Multiphysics (red curves in Figure 2c,d,

and Figure 3c,d), in order to estimate the absorption coefficients for a given set of input data (heat power density profile and sample parameters).

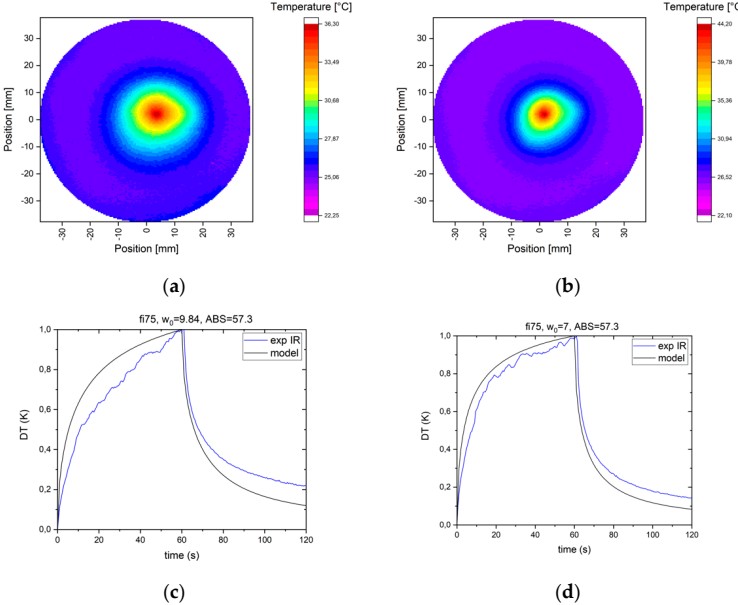

**Figure 2.** Temperature distributions in the $DCH_1$ sample: 2D temperature distributions at 60 s for (**a**) $2w_g = 19.7$ mm and (**b**) $2w_g = 14$ mm. Temperature vs. time for (**c**) $2w_g = 19.7$ mm and (**d**) $2w_g = 14$ mm. The red and blue curves represent the results of the model and experiment, respectively.

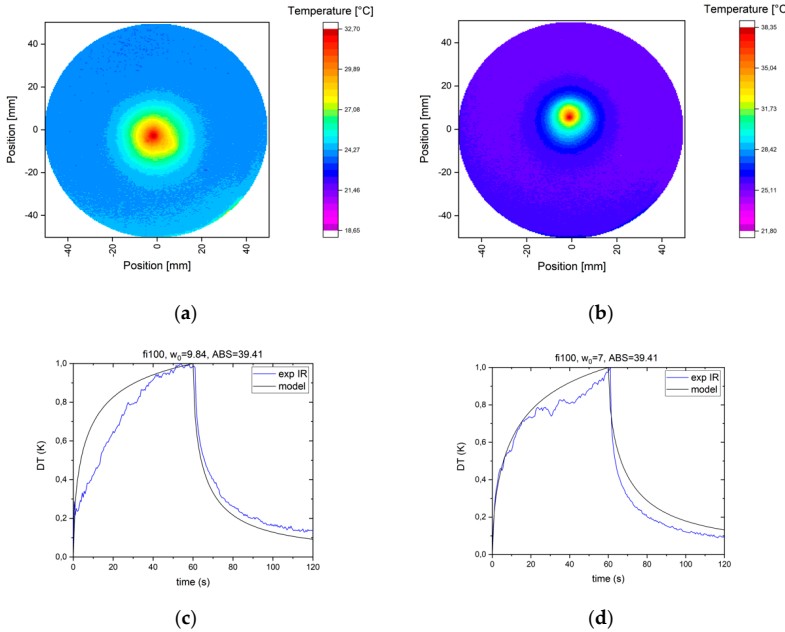

**Figure 3.** Temperature distributions in the $HR_3$ sample: 2D temperature distributions at 60 s for (**a**) $2w_g = 19.7$ mm and (**b**) $2w_g = 14$ mm (**b**). Temperature vs. time for (**c**) $2w_g = 19.7$ mm and (**d**) $2w_g = 14$ mm. The red and blue curves represent the results of the model and experiment, respectively.

For each measurement point (sample corresponding to the column of results in Table 1), we compared two experimental temperature profiles to the model results for an assumed absorption coefficient. In the case of surface absorption losses, it was assumed that, on the given surface, absorption manifests as the "artificial" surface heat source (W/m$^2$) proportional to $A_{surf}$ and the power density. We do not enter here into the physics of absorption and scattering on a thin (a few microns) layer.

The net effect in the form of surface absorption coefficient $A_{surf}$ evidences the quality of technology, i.e., preparing of the surface and coating process.

Assuming that the error in ΔT was 0.3 K, we obtained, for almost all cases (HR1, HR2, HR3, and DCH1), differences of less than 2 ppm between the cases of larger (19.5 mm) and smaller (14 mm) beam diameters. The change of 2 ppm caused changes in ΔT of about 0.5 K.

We are convinced that our approach is valid for a temperature increase of more than 1 K, which corresponds to about 10 ppm of total losses in our specific case for a 10-kW incident power. For the optical elements with losses of less than 10 ppm, this method is problematic. However, we claim that such an approach gives reasonable results for threshold losses of 10 ppm with an uncertainty of about 20% and enables the comparison and validation of several optical technologies for our specific applications.

## 3. Modeling of Transient Thermo-Optic Effects

As demonstrated in Section 2, the maximal temperature increased with the duration of exposition and the absorption loss coefficient. The impact of the transient TOEs on the laser wavefront distortion OPD is quite different for the case using a mirror compared to that using a transmissive element (window or lens).

$$OPD_{mirr} \propto -2\alpha_{CTE}l\Delta T \tag{2}$$

$$OPD_{lens} \propto \left(2\alpha_{CTE}(n-1) + \frac{dn}{dT}\right)l\Delta T \tag{3}$$

Here, $OPD_{mirr}$ is the OPD of a mirror, $OPD_{lens}$ is the OPD of a transmissive element (window or lens), $\alpha_{CTE}$ is the coefficient of thermal expansion, $dn/dT$ is the thermal dispersion of the refractive index $n$, and $\Delta T$ is the temperature increase.

Because of the inhomogeneity of the laser beam profile (for our case, it was a fiber laser source with a close to Gaussian beam profile) and the requirement to model transient, un-stationary effects of the 3D temperature distributions and resulting thermal-elastic deformations of the facets, analytical solutions of the heat equation (see, e.g., [23–26]) cannot be used. Therefore, we decided to use COMSOL Multiphysics for the calculations [27] (Figures 4–7, Table 2). The mirror surface was convex due to the temperature profile, so the resulting wavefront diverged and paraxial thermal lensing occurred, with a negative sign proportional to the thermal expansion coefficient $\alpha_{CTE}$ (dashed red curve in Figure 4). In the case of the transmissive element, due to the convex surfaces of both facets and positive $dn/dT$, the wavefront was concave and paraxial thermal lensing was positive (black curve in Figure 4). Therefore, both TOEs can balance the OPD in principle. In the case of fused silica, $dn/dT = 9.6 \times 10^{-6}$ 1/K and $\alpha_{CTE} = 5.2 \times 10^{-7}$ 1/K; thus, the thermally-induced OPD of the mirror is much weaker compared with that of the lens.

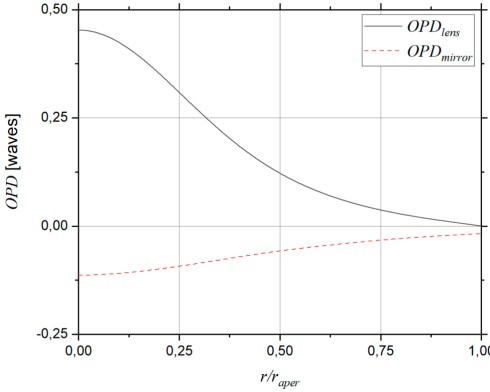

**Figure 4.** Optical path differences (OPDs) of the lens (black curve) and mirror (red dashed curve) vs. the radius for a 9.5-kW beam, $2w_g = 19.7$ mm, and t = 60 s.

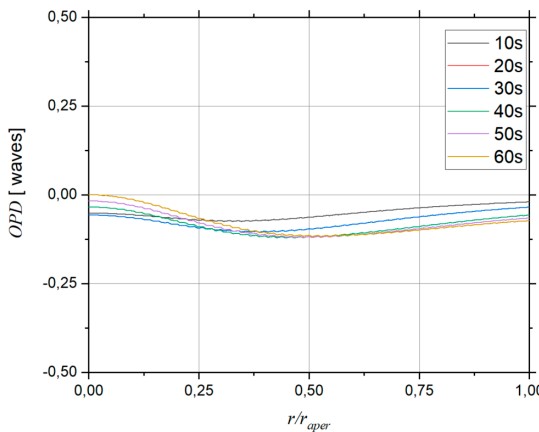

**Figure 5.** OPD of a dynamically compensated train with two components (two mirrors and a compensating lens (CL)) for several exposure durations.

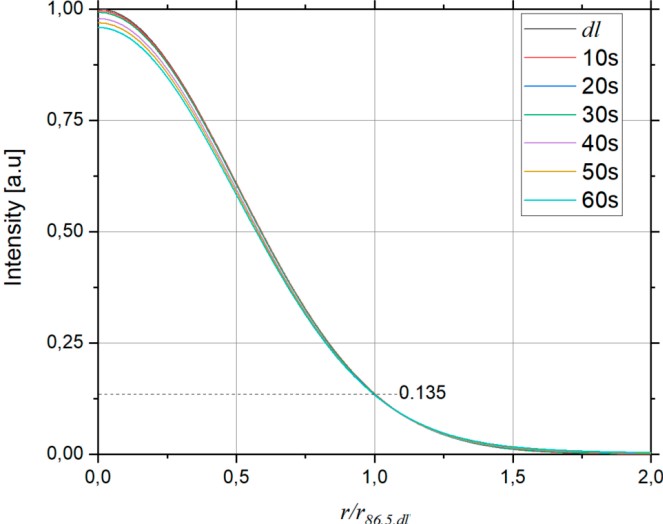

**Figure 6.** Far field intensity of 2 × (CL + 2M) vs. the radius for several exposure durations for the 9.5-kW beam, where $2w_g$ = 19.7 mm.

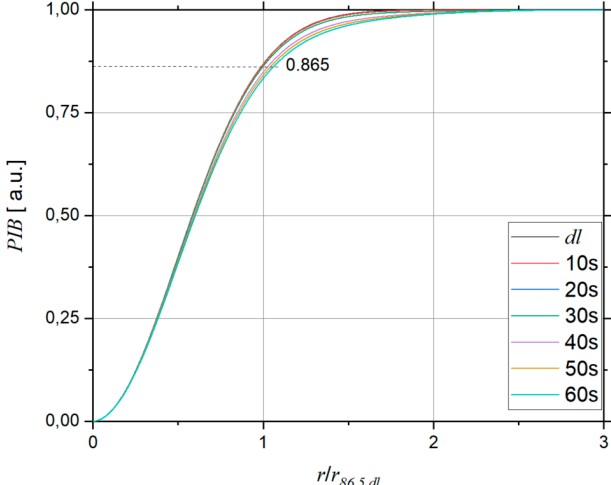

**Figure 7.** Power in the bucket of 2 × (CL + 2M) vs. the radius for several exposure durations for the 9.5-kW beam, where $2w_g$ = 19.7 mm.

**Table 2.** Model results. Optical and laser parameters vs. exposition time: rms (OPD): root mean square of OPD for a given optical system; *SR*: Strehl ratio; $M^2$: beam quality parameter; L: single lens; L + 2M: optical system of a single lens and two mirrors; 2 × (CL + 2M): optical system comprising two groups of compensating lenses (CLs) and two mirrors (M).

| Time (s) | rms (*OPD*) | | | *SR* | | | $M^2$ | | |
|---|---|---|---|---|---|---|---|---|---|
| | L | L + 2M | 2 × (CL + 2M) | L | L + 2M | 2 × (CL + 2M) | L | L + 2M | 2 × (CL + 2M) |
| 10 | 0.027 | 0.007 | 0.016 | 0.982 | 0.999 | 0.997 | 1.029 | 1 | 1 |
| 20 | 0.045 | 0.012 | 0.022 | 0.953 | 0.995 | 0.993 | 1.058 | 1 | 1.014 |
| 30 | 0.058 | 0.017 | 0.022 | 0.926 | 0.991 | 0.993 | 1.101 | 1.014 | 1.022 |
| 40 | 0.068 | 0.022 | 0.026 | 0.903 | 0.985 | 0.979 | 1.145 | 1.029 | 1.043 |
| 50 | 0.075 | 0.026 | 0.03 | 0.884 | 0.98 | 0.969 | 1.159 | 1.029 | 1.058 |
| 60 | 0.082 | 0.03 | 0.036 | 0.868 | 0.975 | 0.959 | 1.188 | 1.029 | 1.072 |

For the absorption coefficient data taken from Table 1, the effective OPD of the train with two components (two mirrors, followed by a compensating lens (CL)) is dynamically balanced with the maximal effective OPD of less than a 0.1 wavelength during 60-s exposition (Figure 5).

To determine the influence of the OPD on the beam quality parameters (Strehl ratio and $M^2$), the following numerical approach was implemented. First, we calculated the intensity distribution $I_{ff,OPD}$ of the aberrated and truncated Gaussian beam in the far field as a Fourier–Bessel transform of the incident Gaussian beam with OPD aberration, as follows:

$$I_{ff,OPD}(\theta) = \left| 2\pi \int_0^{r_{aper}} \exp\left[ik \cdot OPD(\rho) - \left(\rho/w_g\right)^2\right] J_0(k\theta r) r dr \right|^2,\tag{4}$$

and the corresponding intensity distribution $I_{dl}$ of the diffraction limited Gaussian beam truncated at the sample diameter $2r_{aper}$:

$$I_{dl}(\theta) = \left| 2\pi \int_0^{r_{aper}} \exp\left[-\left(\rho/w_g\right)^2\right] J_0(k\theta\rho) \rho d\rho \right|^2,\tag{5}$$

where $w_g$ is the radius of the incident Gaussian beam, $k$ is the wavelength, $\theta$ is the angle in the far field, and *OPD* is the determined experimental/numerical evaluation profile of the OPD (Figures 4 and 5).

Knowing these functions, we can determine the Strehl ratio *SR* as follows:

$$SR = \frac{I_{ff,OPD}(0)}{I_{dl}(0)}.\tag{6}$$

To determine the beam quality parameter $M^2$, we used the definition of power in bucket (PIB) of the beam radius, as follows:

$$M^2 = r_{86.5,OPD}/r_{86.5,dl}.\tag{7}$$

Both radii $r_{86.5,OPD}$ and $r_{86.5,dl}$ were found by numerically solving the following equations of PIB distributions:

$$PIB_{OPD}(r_{86.5,OPD}) = 0.865,\tag{8}$$

$$PIB_{dl}(r_{86.5,dl}) = 0.865,\tag{9}$$

where

$$PIB_{OPD}(r) = \int_0^r I_{ff}(x;OPD)x dx \left/ \int_0^{r_{max}} I_{ff}(x;OPD)x dx \right.,\tag{10}$$

$$PIB_{dl}(r) = \int_0^r I_{dl}(x)xdx \bigg/ \int_0^{r_{max}} I_{dl}(x)xdx. \qquad (11)$$

The far field intensity plots (Figure 6) and PIB curves (Figure 7) were realized based on the OPD data, showing near-perfect compensation of the transient TOEs for the case of the train comprising two special compensating lenses (CLs) and four mirrors (2 × (CL + 2M)). Assuming a perfect Gaussian incident beam, the resulting $M^2$ parameter was less than 1.072 for the entire 60-s exposition duration (Table 2).

## 4. Discussion

The sensitivity limits for the most important loss measurement techniques are presented in Table 3.

**Table 3.** Comparison of loss measurement techniques.

| Loss Measurement Method | Surface Loss Sensitivity (ppm) | Volume Loss Sensitivity (ppm/cm) |
|---|---|---|
| Calorimetry (see, e.g., [28]) | 10–100 | 10–100 |
| Ring down technique [13] | <1 | <1 |
| Cavity mode eigen frequency change [14,17,20,29] | <1 | <1 |
| Thermal deflectometry [12] | <1 | <1 |
| Thermal imaging + model (our approach) | 2 | 5 |

The method of loss measurements proposed in this paper results in the inferior sensitivity compared to the best results obtained during the last few decades (see Table 3). However, as we have shown in part 2 and 3, this approach enables the estimation of residual thermo-optic aberrations and deterioration in the beam quality of the train of optical elements.

The results of the model show that, for a simple lens (Table 2 column 'L'), the transient TOEs are significant and result in a serious reduction in the beam quality ($M^2 = 1.19$). The simple combination of one lens and two mirrors (Table 2, L + 2M) shows partial dynamic compensation of the transient TOEs ($M^2 = 1.03$). Further improvement by employing a special CL via tailored absorption in the coatings enables compensation of a system comprised of two CLs and four mirrors.

This effect was preliminarily confirmed in the experimental characterization of our laser system. First, we did not observe the paraxial thermo-optic-induced focus shift typical for industrial HELs. Therefore, regarding first-order paraxial optics, the system was dynamically compensated. Moreover, the residual higher-order transient TOEs were partly compensated. The observed deterioration in beam quality with power [17,18] was caused by the effects occurring inside the HEL, i.e., heating of gain fibers, transient mode instabilities, and uncompensated transient TOEs occurring in the fiber endcap.

Such an approach can be used in several highly demanding laser optic trains of HELs. Manipulation with substrates and coatings of transmissive element losses (windows, beam splitters, and lenses) and mirrors are attractive for the dynamic self-compensation of high-power laser systems, including the most demanding cases of LIGO.

## 5. Conclusions

Transient 2D temperature distributions in laser optical elements under a 10-kW laser beam exposition were measured and compared to those obtained via numerical modeling in COMSOL Multiphysics. By applying such an experimental/numerical approach, the effective absorption losses in the dielectric layers of typical mirrors and in the volume of the transmissive elements under a high laser power were determined. The layer absorption loss was determined as 20–50 ppm for highly reflective mirrors and less than 10 ppm for antireflective coatings.

The idea of dynamic self-compensation of transient TOEs by a tailored design of the transmissive and reflective elements was proposed and preliminarily verified in a model and experiment. We do not intend to propose any general solution leading to the full, dynamic compensation of thermo-optic effects. Instead of this, we have proposed a solution for a given power density profile and specific

sample geometry and material, in order to find the control parameter (in our case, the value of surface absorption losses of AR coatings of transmissive elements), which gives the best possible compensation of thermally-induced OPDs occurring in a train of examined optical elements. This is only a proposal of basic idea, and how to change the effective absorption of coatings is a task or challenge for optical technology.

The numerical model of this concept for two dynamically compensated optical trains, with one comprising a single lens and two mirrors (L + 2M) and the second one comprising two special compensated lenses and four mirrors (2 × (CL + 2M)), has been presented.

**Author Contributions:** Conceptualization, J.J., L.G., and K.K.; methodology, J.J. and L.G.; software, L.G., P.G., and M.D.; formal analysis and modeling, P.G. and J.J.; investigation, L.G., P.G., P.T., and M.D.; optical technology, J.F., M.S., and K.K.; resources, J.B., J.F., K.F., M.M., M.S., T.O., T.S., and K.K.; writing—original draft preparation, J.J.; writing—review and editing, L.G. and P.G.; supervision, J.J. All authors have read and agreed to the published version of the manuscript.

**Funding:** The work was financed in the framework of the strategic program DOB-1-6/1/PS/2014 by the National Centre for Research and Development of Poland.

**Acknowledgments:** We thank Henryk Madura from the Institute of Optoelectronics MUT for consultation on thermal measurements.

**Conflicts of Interest:** The authors declare no conflict of interest.

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
