# Peer review of "Characterization of Absorption Losses and Transient Thermo-Optic Effects in a High-Power Laser System"

_photonics, doi:10.3390/photonics7040094_

Round 1

Reviewer 1 Report

The authors present the study of the performance of the optical system (and its parts) delivering the 10 kW cw laser beam produced by the commercial fiber laser system to the target. It is stated, taht even if the laser is operated in few seconds „pulses“ the thermal effects induced in the optical components such as lenses and mirrors is decreasing the beam quality significantly. The authors observed the positive effect of the combination of the transmissive and reflective components to the beam distortion by the thermal lensing effect. The authors explained this positive inbfluence by the fact that the typical transmissive optical component produces positive thermal lens (if dn/dT > 0) while the reflective elements typically produces negative thermal lensing effect thus providing the compensation of the contribution of the transmissive components. The partial thermal lens compensation by the combination of trnsmissive and reflective optical components was demonstrated on the laser system powered by commercial 10 kW fiber laser system.

From my point of view the most important result of the manuscript would be the possible mutual compensation of the thermal lens of transmissive and reflecting optical components. The fact that these types of elements are producing the oposit thermal lens is known, but in my opinion it has never been detailed studied from the point of view of mutual compensation of these effects. Nevertheless, the results of the absorption measurement and also the compensation should be justified by more exact approach. In the present form, the results are, in my opinion, not exactly justified. The detailed explanation is provided below. For this reason I am not convinced about the suitability of the manuscript in the current form for publication in Photonics journal. The most important comments (the first number is the number of the line) are the following:

  1. 48: there is a value of the volume absortion < 5 ppm/cm but there is no information about the source of this value. Is that taken from some reference or it is the original measurement?
  2. 49: „...other advantageous thermo-mechanical parameters“ what kind of parameters and are these really so advantageous in the case of fused silica? The information is very vague and the authors should be more specific.
  3. 68: „The main goal of the experiments was to determine effective absorption coefficients...“ It is not clear to me how was the absorption coeeficien determined, especially with ppm precision. Was this coefficient calculated from the matching of the IR camera temperature measurement with COMSOL model? In that case the measurement should be described in more details. Especially, the authors should address the following: what was thw real precision of the temperature measurement by IR camera, the presented precision 0.025 K seems unrealistic e.g. how was the emissivity of the surface evaluated so precisely. How the authors removed the Fresnel loss on the surfaces from the calculation? How were the boundary conditions for the calculation determined so precisely, there is not even mention about these. However, the boundary conditions can strongly influence the results and it is not easy to specify them precisely. How precisely are the material parameters (thermal, mechanical, and optical) known, there are usually no references to the used values. It should be also noted that all these parameters are temperature-dependent which is the fact, which was not taken into considerationt.
  4. 90 Table 1: What is exactly ment by „surface absorption loss coefficient“? It is not explained. Is that some average value of absortion coefficient for surface multilayers? If yes, how is it defined?
  5. 2 and 3: The axes labels are quite small and it is difficult to read them. There is no label for figs a) and b) at least colorbar would be beneficial. Otherwise it is just a colored circle. Moreover the tempertaure results seems to me just as relative measurement (scaled from 0 to 1)  and the temperature obtained from the calculation was adjusted by some coefficient to the same maximal value which was obtained from the measurement. Such approach however must be justified if these values were used for the calculation of the absorption coefficient.
  6. 106: How the usage of smaller beam verifies the results? The evaluation has been done by exactly the same approach as for the wide beam.  
  7. 114: For the reasons described in the points 3-6 I have doubts about the precision of the measurement provided by authors (few ppm or 20 %). It is generaly difficult to measure such a low absorption coefficient especially with such precision. To the best of my knowledge the only method which is able to provide such precision is photothermal interferometry (which was used for the characterization of LIGO components mentioned in the manuscript). The authors should justify their results more exactly.
  8. 124: what is ment by in-homogeneity of the incident beam? What beam shape was used for the COMSOL calculation? The profile taken from experiment? From the figures it seems like almost perfect gaussian beam. However, there exist the analytical solution for gaussian beam. But just for the strictly radial thermal flux and for the bulking of the faces neglected (plain strain approximation) or with axially homogeneous het load (plain stress approximation). The main reason for the non-existence of analytical solution in this case would be probably the axial heat flux given by the presence of axial non-homogenity of heat load and heat exchange of the faces of optical elements with an ambient. Could authors provide a more detailed explanation to this?
  9. It seems to me, that the OPD is just calculated. It is a pity, that there is no OPD measurement, because for the validation of the results it is far more exact than the surface temperature measurement, while the OPD can be measure much more precisely.

Reviewer 2 Report

Thermal-optical effects on optical elements in a high power laser system is studied in this manuscript. Different with the well investigated thermal effect in the gain medium of laser, this work focuses on the high power influence on optical elements, such as lenses and mirrors. The OPDs of elements under different illumination cases were measured. And the idea of OPD compensation by different combination of elements was given.

This manuscript is of value to be published in Photonics.

While, several questions as below should be considered by the authors before it be accepted.

  1. In line 42, “we have developed a the laboratory” and “based on commercial a” The two “a” may have problems in grammar; such problems in grammar should also be corrected.
  2. It’s better to provide the IR figures detected by the IRC, to make a comparison with the 2D temperature distributions, if possible;
  3. What’s the absolute value of temperature in figure 2&3. And a color bar should be added to fig.2&3(a) and (b);
  4. Does the curvature of radius of the surface of a lens (or the focal length) have influences on the OPD and compensation condition? (under the same illumination case)

Reviewer 3 Report

Comments to the manuscript entitled "Characterization of Absorption Losses and Transient Thermal-Optic Effects in High-Power Laser System"

I wish to start by saying that joint studies (experimental and simulations)  are extremely important in terms of all practical applications; therefore, every study using this cooperative approach is always welcome to the scientific community.

The content of the manuscript is of interest, in particular, for the high energy lasers field. The authors show a broad and in-depth theoretical knowledge regarding the design procedure and the analysis of the results. As the authors correctly mentioned, the concept of dynamic self-compensation of thermal-optical effects has been studied in the past two decades. However, their approach of using two dynamically compensated optical trains seems to be less problematic and ingenious, while compared with other methods (I wish to stress the verb "seem").

As I mentioned before, the authors revealed in-depth theoretical knowledge of this subject, and I cannot find any significant corrections to be made in the manuscript. However, I have found a significant drawback in the submitted manuscript, related to the lack of results comparison with other previous works from different research groups. I strongly suggest that the authors include a table in the "Results" or "Discussion" section for a comparison purpose and the respective discussion. Such a discussion of results will improve the submitted manuscript's scientific significance and enhance the reader's experience.

That said, I will recommend the manuscript's acceptance.

Round 2

Reviewer 1 Report

I would like to thank authors for their effort which they paid to the explanations and modifications of their manuscript. In my opinion the manuscript can be considered for publication in Photonics journal in the its current form.